# FAST BINARY FUNCTIONAL SEARCH ON GRAPH

## ABSTRACT

Large-scale search is an essential task in modern information systems. Numerous learning based models are proposed to capture semantic level similarity measures for searching or ranking. However, these measures are usually complicated and beyond metric distances. As Approximate Nearest Neighbor Search (ANNS) techniques have specifications on metric distances, efficient searching by advanced measures is still an open question. In this paper, we formulate large-scale search as a general task, Optimal Binary Functional Search (OBFS), which contains ANNS as special cases. We analyze existing OBFS methods' limitations and explain they are not applicable for complicated searching measures. We propose a flexible graph based solution for OBFS, Search on L2 Graph (SL2G). SL2G can achieve approximate optimal, with accessible conditions. Experiments demonstrate SL2G's efficiency in searching by advanced matching measures (i.e., Neural Network based measures).

## 1 INTRODUCTION

Approximate Nearest Neighbor Search (ANNS), or Similarity Search, has been a classical data science task for decades. In the ANNS paradigm, each given query is compared with a subset instead of the whole dataset, which reduces time complexity significantly and show powerful performance in Search Engines and Recommender Systems. To obtain the subset with a high recall rate, specific index structures are proposed for dense continuous vectors (Charikar, 2002; Jegou et al., 2008) or high dimensional sparse data (Broder, 1997; Li et al., 2012). For the dense case, which is our focus in this paper, hashing based algorithms (Gionis et al., 1999; Charikar, 2002; Weiss et al., 2009), quantization-based methods (Jegou et al., 2008; 2011; Ge et al., 2013; Wu et al., 2017) and graph based approaches (Hajebi et al., 2011; Malkov et al., 2014; Wu et al., 2014; Malkov & Yashunin, 2016; Fu et al., 2017) are applied. Particularly, graph based methods show promise in improving ANNS performance (Aumüller et al., 2017). Besides, graph based index structures have the flexibility that user can define any matching function as similarity measure in graph construction and searching.

However, most existing graph based ANNS methods concentrate on searching by metric distances, such as $\ell_2$ distance or angular distance. Efficient searching beyond metric distances is considered hard for ANNS. Only a few simple non-metric measures were studied in literature, such as Maximum Inner Product Search (MIPS) (Shrivastava & Li, 2014), Max-Kernel Search (Curtin et al., 2013) and searching by Bregman divergence (Cayton, 2008). For more complicated matching measures, efficient searching is still an open question. Development of machine learning techniques aggravates this problem by increasing matching measures' complexity. Recently, learning based ranking/matching models were continuously proposed to capture semantic level relevance. Among various learning paradigms, **Matching Function Learning** becomes popular these years (He et al., 2017; Xiong et al., 2017; Tay et al., 2018; Dai et al., 2018; Xu et al., 2018). It assumes a mapping exists between query and document (or user and item). Therefore, the goal of Matching Function Learning is to learn the mapping by tunable models, such as Neural Networks. Matching Function Learning is widely adopted in academic works for its significant performance. However, as the learned matching functions are usually too complicated than metric distances, efficient searching and on-line model prediction remains scarce. Thus, the application of Matching Function Learning approaches in real searching or recommendation systems is still constraining.

In this paper, we introduce a general task by formulating a Optimal Binary Functional Search (OBFS) problem. ANNS and MIPS problems can be considered as its special cases. Under the

definition of OBFS, performing Search on Graph methods can be summarized as a common methodology, Search on Binary Functional Graph (SBFG): constructing the (approximate) Delaunay graph with respect to binary functional $f$ for all indexing data and then greedy searching most relevant vertices on the graph for each query $q$. We validate SBFG's outperforming efficiency theoretically on metric distances. However, constructing or approximately constructing Delaunay graphs for complicated binary functionals is extremely hard. It is difficult to extend SBFG to more general OBFS problems, such as OBFS with Neural Network as binary functionals.

To overcome the limitations of SBFG, we propose a new graph based searching methodology, Search on L2 Graph (SL2G). Different from SBFG, SL2G constructs the index graph by $\ell_2$ distance among searching data, no matter which binary functional $f$ we work on. In the searching phase, SL2G works similarly with SBFG: greedy search by the focus binary functional $f$. Paradigms of SL2G breaks the symmetric limitation of SBFG. As analyzed in the theoretical part, SL2G approximates the gradient descent in Euclidean space. Even if $f$ is non-convex, SL2G can reach an approximate local optimal, based on some affordable assumptions.

In experiments, we test the proposed methodology, SL2G, by two Neural Network based binary functions. As will be seen, as SL2G is designed for general OBFS problem, it works consistently well on different matching measures (i.e., binary functionals). But the traditional methodology, SBFG, works badly on complicated matching measures, especially for asymmetric measures.

In summary, the contributions of this paper are as below:

- Extend ANNS to a more general task, Optimal Binary Functional Search (OBFS). Under OBFS, more advanced matching measures can be considered in large-scale search.

- Theoretically explain why existing Search on Graph methods work well on metric similarity measures (e.g., $\ell_2$ distance) and specify their limitations when extending to complicated searching measures.

- Propose a solution, SL2G, to tackle the OBFS problem. SL2G breaks the limitations of existing Search on Graph methods for OBFS and is applicable for any complex searching measures (e.g., Neural Network models). Theoretical analysis for SL2G is provided.

## 2 BACKGROUND OF SEARCH ON GRAPHS

Graph structures provide a natural way to partition a high dimensional continuous space into discrete regions. Theoretically, formulation of Voronoi Diagram and its dual graph, Delaunay Graph (Fortune, 1995; Aurenhammer, 1991) provide the foundation for Nearest Neighbor Search (NNS) on graph. If the space is partitioned via Voronoi Diagram, nearest neighbors of given queries can be returned by Delaunay Graph (or Delaunay triangulation) with exactness. Generally, in order to preserve the preciseness in NNS, the network must contain the Delaunay Graph as its subgraph. However, as Delaunay Graphs quickly become fully connected at high dimensionality, efficiency of NNS on Delaunay Graphs reduces dramatically (Harwood & Drummond, 2016). Moreover, for applications of NNS, finding exact nearest neighbors can be substituted by Approximate NNS. Therefore, support for whole and exact Delaunay Graph is not required in ANNS. Instead, alternative graphs are proposed to approximate their performance with affordable time complexity.

$k$ Nearest Neighbor ($k$NN) graph is claimed to be an approximation of Delaunay Graph (Fu et al., 2017; Hajebi et al., 2011). For each vertex, $k$NN graph connects vertices within top $k$-th smallest distances to it. Then greedy search is performed on $k$NN graph to return nearest neighbors. In experiments, $k$NN graphs show effectiveness in ANNS on high dimensional real world spaces. Hajebi et al. (2011), Fu & Cai (2016) and Jin et al. (2014) use hashing and randomized KD-trees to provide better starting positions of greedy search on graph and show promising performance in ANNS.

Inspired by the small world phenomenon, Navigable Small World Network (NSWN) is introduced (Kleinberg, 2000) and shown their potentials in ANNS. Malkov et al. (2014) propose a Navigable Small World (NSW) graph that approximates both NSWN and Delaunay Graphs simultaneously. NSW consists short-range links and long-range links as its edges. Short-range links are approximations to Delaunay Graphs, which ensure that approximate nearest neighbors will be returned via greedy search. Long-range links are regarded as shortcuts, which are responsible for navigation between small world clusters. The existence of long-range links help logarithmic scaling of greedy

search on graphs. Thus NSW demonstrates better performance than $k$NN graph at high dimensionality. However, NSW's efficiency is constraining as degree of the graph is too high and connectivity problems may exist in graph. To tackle this issue, Malkov and Yashunin's following work (Malkov & Yashunin, 2016) propose Hierarchical NSW (HNSW), an extended version of NSW. HNSW adopts edge selection strategy from Relative Neighborhood Graphs (RNG) to reduce degree of the NSW graph. Meanwhile, a hierarchical structure is used in HNSW to perform multi-scale hopping. This "zoom-out" and "zoom-in" effect offers significantly better performance.

Although there are few theoretical literature on proving whether approximately constructed graphs contain Delaunay Graph as subgraph, both $k$NN and NSW graphs perform Delaunay Graph's proprieties empirically. Actually, graph based ANNS algorithms significantly outperforms hashing based (Gionis et al., 1999; Lv et al., 2007) and quantization based ANNS methods (Jegou et al., 2008; 2011) in well-known benchmarks (Aumüller et al., 2017).

## 3 BINARY FUNCTIONAL SEARCH ON GRAPH

In this section, we extend ANNS problem to a broader setting. Before introducing the proposed method for the new defined problem, we will first analyze theoretical foundations of existing Search on Graph methods and their limitations.

### 3.1 DEFINITIONS AND NOTATIONS

Although Search on Graph methods claimed that there are no constraints on measures defined in graph constructing and searching (Hajebi et al., 2011; Malkov & Yashunin, 2016), most existing ANNS works mainly focus on searching by some metric distances. It was shown that graph based methods are outperforming on a few non-metric measures in empirical experiments (Malkov & Yashunin, 2016), but more potential searching measures remain to be discussed. Meanwhile, systemic theoretical analysis has few been conducted on the measure variation.

In this paper, we formulate the searching task in a more general way, **Optimal Binary Functional Search** (**OBFS**). It will be shown that ANNS and MIPS are special cases of OBFS.

**Definition 1.** *(OBFS) Let $X$ and $Y$ be subsets of Euclidean spaces (possibly with different dimensions), given a data set $S = \{x_1, \ldots, x_n\} \subset X$ and a continuous binary functional, $f : X \times Y \to \mathbb{R}$, OBFS aims to find*

$$\arg\max_{x_i \in S} f(x_i, q), \quad \text{for } q \in Y. \tag{1}$$

Let us consider $X \subset \mathbb{R}^d$ and $Y \subset \mathbb{R}^k$. Note that we consider subsets of Euclidean space because some functions' domain is not the whole space, for example, cosine is undefined at origin. Take the recommendation problem as an example. We consider items as searching data $X$ and the user set as queries $Y$. Matching function is specified or learned by Binary functional $f$, which takes a pair of item and user as input. The output of $f$ is the ranking score for recommendation. $\ell_2$ distance, angular distance and inner product are special cases of binary functionals when $d = k$. Since Search on Graph methods are based on Delaunay graphs, we also introduce related definitions here.

**Definition 2.** *The Voronoi cell $R_i$ with respect to $f$ and $x_i$ is the set*

$$R_i := \{q \in Y : f(x_i, q) \geq f(x_j, q) \text{ for } j \neq i\}. \tag{2}$$

**Definition 3.** *The Delaunay graph $G$ with respect to $f$ and $S$ is an undirected graph with vertices $S$ satisfies $\{x_i, x_j\} \in G$ if and only if $R_i \cap R_j \neq \emptyset$.*

That is two data points in the Delaunay graph will be connected if the Voronoi cell is adjacent to each other. Here, adjacency means their boundary has nonempty intersection. It is worth noting that, the Voronoi diagram is built on $Y$, while the Delaunay graph is constructed with vertices $S$ on $X$. For an undirected graph $G$, we say $\{x_i, x_j\} \in G$ if there is an edge between $x_i$ and $x_j$. We refer Delaunay graphs w.r.t. any $f$ on some data set $S$ as **Binary Functional Graph**.

**Example 1.** *Let $f : \mathbb{R}^d \times \mathbb{R}^d \to \mathbb{R}$ be negative $\ell_2$ distance, i,e., $f(x, y) = -\|x - y\|$, then the Delaunay graph defined in Definition 3 coincides with classical definition, for instance, the one in Lee & Schachter (1980).*

Notations used in this paper are summarized in Table 1.

Table 1: Notations Used in This Paper.

| | |
|---|---|
| $f : X \times Y \to \mathbb{R}$ | A continuous binary functional, $X \subset \mathbb{R}^d$ and $Y \subset \mathbb{R}^k$ |
| $R_i$ | A Voronoi cell with respect to data point $x_i$ |
| $\|\cdot\|$ | $\ell_2$ norm in Euclidean space |
| $B$ | An open ball in Euclidean space |
| $\partial B$ | the boundary of $B$ |
| $\overline{B}$ | The closure of $B$, $\overline{B} = B \cup \partial B$ |
| $f_q(\cdot)$ | For $q \in Y$, $f_q(x) = f(x, q)$ |
| $f_q^{-1}(A)$ | $\{x \in X : f_q(x) \in A\}$, the preimage of $A \in \mathbb{R}$. |
| $f_q^{-1}(a)$ | $\{x \in X : f_q(x) = a\} = f_q^{-1}(\{a\})$, a level set of $f_q$. |

## 3.2 OBFS on Binary Functional Graph

For the OBFS problem, previous graph based works focus on special cases as mentioned above. The common methodology behind these works can be summarized as, **Search on Binary Functional Graph** (**SBFG**). That is constructing the (approximate) Delaunay graph with respect to binary functional $f$ for all indexing data $S$ and then greedy searching most relevant vertices on the graph for each query $q$ by $f$. As shown in empirical results by previous papers, SBFG works well for measures such as $\ell_2$ distance and angular distances. Here we provide theoretical analysis for the reason why SBFG works for some cases of OBFS.

**Theorem 1.** *Suppose a continuous binary functional $f$ satisfies the Voronoi cells $R_i$ with respect to any finite subset are connected on $Y$, and $G$ is the Delaunay graph with respect to $f$ on some $S$, then for $q \in Y \subset \mathbb{R}^k$, a local maximum in the greedy search on $G$, that is, $x_i \in S$ satisfies*

$$f(x_i, q) \geq \max_{x \in N(x_i)} f(x, q), \quad \text{where} \quad N(x_i) = \{x \in S : \{x_i, x\} \in G\}, \tag{3}$$

*is a global maximum.*

That means for an arbitrary query, a greedy search on Delaunay graph with any initial point can find the nearest neighbor of the query. The proof can be found in Appendix A.

**Corollary 1.** *Under Theorem 1's assumption, if we replace the graph $G$ by a graph that contains Delaunay graph as a subgraph, Theorem 1 holds.*

Assuming we can build any Delaunay graphs in Definition 3, Theorem 1 implies that SBFG works well in some cases such as:

- $X = Y = \mathbb{R}^d$, $f(x, y) = -\|x - y\|$. (Example 1)

- $X = Y = \mathbb{R}^d \backslash \{0\}$, $f(x, y) = \cos(x, y) = \frac{x^\top y}{\|x\|\|y\|}$.

- $X = \mathbb{R}^d, Y = \mathbb{R}^k \backslash \{0\}$, $f(x, y) = x^\top A y$ for some $A \in \mathbb{R}^{d \times k}$. In particular, if $d = k$ and $A = I_d$, then this OBFS problem is exactly the MIPS problem.

However, how to construct the Delaunay graph algorithmically w.r.t. $f$ is challenging. Some algorithms have been developed to build Delaunay graph w.r.t. $\ell_2$ distance (Lee & Schachter, 1980), (Fortune, 1995),(Cignoni et al., 1998). These algorithms could be extended to construct approximate Delaunay graphs w.r.t. other simple measures beyond $\ell_2$, such as angular distance or Inner Product. But it would be difficult to extend them to complicated binary functionals such as asymmetric ones. Algorithms for Delaunay graph w.r.t. any asymmetric binary functionals have never been investigated yet. We will bypass building Delaunay graphs w.r.t. complicated binary functionals and propose a new algorithm in the next section, which is only based on Delaunay graphs w.r.t. $\ell_2$ distance.

## 3.3 OBFS on L2 Graph

As analyzed above, building Delaunay graphs w.r.t. $f$ is algorithmically difficult for most of binary functionals besides some easy cases, such as $\ell_2$ distance. So we proposed a new methodology,

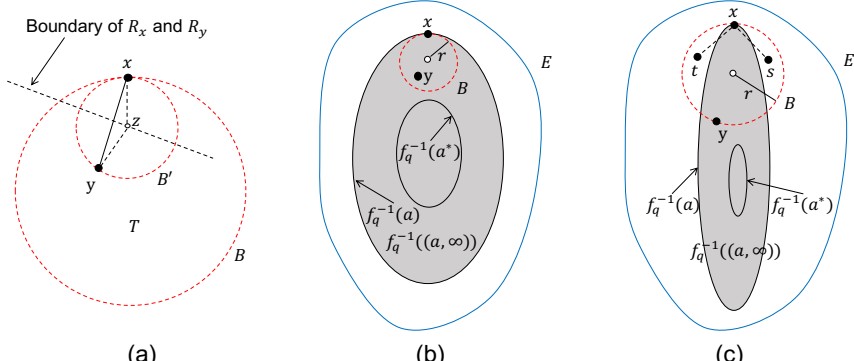

Figure 1: (a) is the schematic diagram for the proof of Lemma 1. (b) and (c) are for the proof of Theorem 2. (b) is for the assumptions of the theorem are satisfied. While if the shape $f$ is strange and the data is not dense enough, the theorem will not hold as shown in (c). In (c), there are two data points $t$ and $s$ may take place $y$ and connect with $x$. Then the optimization will stop at $x$.

**Search on L2 Graph** (**SL2G**), for the OBFS problem instead of SBFG. The basic idea of SL2G is: no matter what the given any binary functional $f$ is, we construct a Delaunay graph (or an approximate one) w.r.t. $\ell_2$ distance (which is defined on searching data $X$ and independent of queries) in the indexing step and then perform greedy search on this graph by the binary functional $f$ in the searching step. SL2G only construct Delaunay graphs w.r.t. $\ell_2$ distance but not the focus functional $f$. As building Delaunay graph w.r.t. $\ell_2$ distance is with many previous algorithms, SL2G is applicable for any general binary functionals. If $f$ is the negative $\ell_2$ distance as in Example 1, SBFG and SL2G are equivalent.

The key factor of SL2G is not only the feasibility of Delaunay graph building but also the effectiveness in solving the OBFS problem. We will show that when the (finite) dataset $S$ is dense enough in certain region, the performance of greedy search on $\ell_2$ Delaunay graph is similar to optimize Equation 1 by gradient descent in Euclidean space, based on some affordable assumptions.

**Lemma 1.** *Let $B$ be an open ball in $\mathbb{R}^d$, $x \in \partial B$ (i.e., $x$ is on the boundary of $B$), and $T := S \cap B \neq \emptyset$, then the $\ell_2$ Delaunay graph with respect to $S$ connects $x$ with at least one point in $T$.*

*Proof.* We consider internally tangent spheres $\partial B'$ of $\partial B$ that has common external intersect at $x$. We pick a $B'$ such that there exists $y \in S$ satisfying $y \in \partial B'$ (i.e., $y$ is on the boundary of $B'$) and $T \cap B' = \emptyset$. Then the nearest data points from $S$ to the center $z$ of $B'$ are $x$ and $y$. The Voronoi cells $R_x$ of $x$ and $R_y$ of $y$ intersects at $z$, so $x$ and $y$ are connected in the Delaunay Graph. □

For fixed $q \in Y$, let $f_q(x) = f(x, q)$. Note that $f_q$ could be non-concave functions and optimization by gradient descent may stop at local optimal. To simply the discussion, we only consider a convex set for one local optimal of $f_q$ if $f_q$ is non-concave.

**Theorem 2.** *Let $f_q$ be a concave function defined on a compact convex set $E \in \mathbb{R}^d$. We assume there exists $r > 0$ such that the followings are true:*

(a) *There exist $a^* < \max_{x \in E} f_q(x)$, for every $a \leq a^*$ and $x \in f_q^{-1}(a)$, there exists a d-dimensional open ball $B$ with radius $r$ such that $\partial B \cap f_q^{-1}(a) = x$ and $B \subset f_q^{-1}((a, \infty))$.*

(b) *For every open $r$-ball $B \subset E$, we have $B \cap S \neq \emptyset$.*

*Then the greedy search on the $\ell_2$ Delaunay graph w.r.t. $S$ achieves value at least $a^*$.*

*Proof.* Let us consider a $x \in S$ such that $f_q(x) \leq a^*$, and let $B$ be the $r$-ball defined in the theorem. There exists at least one point in $S$ dropped on $B$. By Lemma 1, $x$ connects with at least one point in $B$, which has higher evaluation in $f_q$. The updates will stop only if it achieves a value larger than $a^*$, which is the approximation of local optimal. □

As shown in Figure 1 (b) and (c), only when the shape of the binary functional $f$ is too strange and the data points are too sparse in the specific area, the theorem will not hold. For example, in Figure 1 (c), there are two data points $t$ and $s$ which are closer to $x$ in $f_q$ than $y$. They may take place $y$ and connect with $x$. Without the edge between $x$ and $y$, the optimization will not go further towards level set $f_q^{-1}(a^*)$ and will stop at $x$. It can be proven that assumption (b) listed in Theorem 2 is not difficult to meet. We put the proof in Appendix B because of the limit space. Assumption (a) holds when the radius of curvature at every point on the manifold $f_q^{-1}(a)$ is at least $r$. It is worth noting that, searching for such a local optimum only require edges on $\ell_2$ Delaunay graph with length less than $r$, so a complete $\ell_2$ Delaunay graph is not necessary. We can also weaken concavity of $f_q$ to quasi-concavity as long as the assumptions still hold.

## 3.4 Implementations of SL2G

In practice, constructing a perfect Delaunay graph is usually time consuming or computationally infeasible. As summarized in Section 2, there are some previous works tried to construct approximate Delaunay graphs. To meet the efficient searching purpose, the vertex degrees of these graphs are often restricted to pretty low amounts. That will sacrifices the properties of Delaunay graph. So there is no theoretical guarantees that Search on Graph methods will return exact optimal results via Delaunay graphs approximation. According to empirical experiments, the effectiveness of approximate Delaunay graphs w.r.t. $\ell_2$ distance was proven by previous works (Hajebi et al., 2011; Malkov & Yashunin, 2016). But for general binary functional $f$'s beyond $\ell_2$ distance, such as asymmetric $f$'s or Neural Network based $f$'s, approximate and efficient Delaunay graph constructing is still an open problem. Nevertheless, SL2G only constructs Delaunay graphs w.r.t. $\ell_2$ distance, which break through the format constraints of searching measure $f$.

---

**Algorithm 1** L2 Graph Construction for SL2G

1: **input:** binary functional $f$, data set $S$ and maximum vertex degree $M$, search depth $N$.
2: Initialize graph $G = \emptyset$
3: **for** each $x_i$ in $S$ **do**
4:     Greedy search $N$ vertices $\{x_j\}$ on $G$ with largest values with $x_i$ in negative $\ell_2$ distance in descending order.
5:     $C = \emptyset$.
6:     **for** j = 1 to N **do**
7:         **if** $\|x_i - x_j\| < \min_{x \in C} \|x - x_j\|$ **then** $C = C \cup \{x_j\}$, add edge $\{x_i, x_j\}$ to $G$.
8:         **end if**
9:         **if** $|C| = M$ **then** break
10:         **end if**
11:     **end for**
12: **end for**
13: **output:** graph $G$

---

In this paper, we take HNSW Malkov & Yashunin (2016) as the implementation for studying SL2G. Reasons are: (1) HNSW is the state-of-the-art Search on Graph method and outperforms other ANNS algorithms on benchmarks (Aumüller et al., 2017). (2) HNSW approximates both Small World graph and Delaunay graph. The long range edges of Small World graphs would be helpful in avoiding local optimal, which SL2G may suffer from. The graph construction algorithm for SL2G is shown in Algorithm 1. The edge selection method represented in Line 7 is from the original HNSW method, which was shown that it improves the performance greatly in the trade-off Recall vs. Time. No matter what binary functional $f$ we focus on, SL2G constructs graphs by negative $\ell_2$ distance as shown in Line 4. The greedy search algorithm of SL2G is similar with the original HNSW but replacing the metric searching measures by the focus binary functional $f$, which is defined on data-query pair in searching. The graph construction algorithm for SBFG and the greedy search algorithm for both SBFG and SL2G can be found in Appendix C.

# 4 EXPERIMENTS

In this section, we evaluate the performance of SL2G (working with HNSW) for two Neural Network based relevant measures for recommendation. One is symmetric and the other is asymmetric.

## 4.1 DATASETS AND EXPERIMENTAL SETTINGS

For experimental datasets, we choose two widely used datasets for recommendation: **Yelp**[1] and Amazon Movie (**Amovie**) [2] . For Yelp, we did not further filter it for it was processed before. It contains 25677 users, 25815 items and 731670 ratings. For Amovie, we filtered the dataset in the way that users with at least 30 interactions are retained. At last Amovie contains 7748 users, 104708 items and 746397 ratings. The item amount in these two datasets is much bigger than that in others, which is more appropriate for searching efficiency exploration.

To generate evaluating labels, we calculate most relevant items for each user by the corresponding binary functional $f$, which will be introduced later. Then experiments on Top-1, 10, 50 and 100 labels will be recorded. Although we use datasets from the recommendation domain, we will not test the recommendation precision by true labels (i.e., previous user ratings). We only focus on searching efficiency of various searching algorithms. For the evaluating measure, we adopt the trade-off **Recall vs. Time**. Recall vs. Time reports the amount of queries an algorithm can process per second at each recall rate level.

HNSW has three parameters: $M$, $efConstruction$ and $efSearch$, which control the degrees of vertices and the number of search attempts. To make fair comparison, we vary these parameters over a fine grid. For each algorithm in each experiment, we will have multiple points scattered on the plane. To plot curves, we first find out the best result, $max_x$, along x-axis (i.e., Recall). Then 100 buckets are produced by splitting the range from 0 to $max_x$ evenly. For each bucket, the best result along y-axis (i.e., the biggest amount of queries per second ) is chosen. If there is no data points in the current bucket, the closest result on the left bucket will be chosen instead. In this way, we shall have 100 pairs of data for drawing curves.

Experiments were performed on a 2X 3.00 GHz 8-core i7-5960X CPU server with 32GB memory.

## 4.2 NEURAL NETWORK BASED BINARY FUNCTIONALS

To evaluate the performance of SL2G on complicated binary functionals, we leverage a state-of-the-art Neural Network based recommendation method, **MLP**, which was introduced in (He et al., 2017). The original MLP model concatenates user latent vectors and item latent vectors before going through the Multi-Layer Perceptron network (referred as **MLP-Concate**). The concatenating operation is asymmetric. We design one more variant of MLP with symmetric operations on the input vectors: **MLP-Em-Sum**. MLP-Em-Sum transforms two kinds of vectors into a common space by an additional embedding layer, before the merge operation. In this way, user vectors and item vectors will lie on the same manifold on which element-wise sum will be operated. After training, we obtain NN models with fixed weights. These fixed weight models can be considered as binary functional $f$'s. For each pair of input vectors, (user, item), the binary functional $f$ will output a real number. These model based binary functionals are referred as $f_{\text{MLP-Concate}}$ and $f_{\text{MLP-Em-Sum}}$. Note that, for both models, only the part above the embedding vectors of the network (i.e., the Multi-Layer Perceptron network) is regarded as the matching functional. The part from raw data to the embedding layer does not belong to the matching function. Since this is the first work for OBFS with Neural Network based searching measures and there are few previous comparable algorithms. SL2G is the only solution till now (referred as **HNSW-SL2G**). For implementation, after parameters of each model being learned, we re-implement and integrate the model into the HNSW framework as the searching measure. The dimensions of input vectors for MLP-Em-Sum are set as 64 and for MLP-Concate are set as 32 (after concatenating, the vectors become 64-dimensional too).

We choose SBFG on HNSW as the baseline here (referred as **HNSW-SBFG**). We adjust SBFG for complicated binary functionals as below: we input a pair of searching data to the binary functional forcibly as $f(x, x)$. Then we use this $f(x, x)$ as relevance measure to construct the index graph

---

[1]https://www.Yelp.com/dataset/challenge

[2]http://jmcauley.ucsd.edu/data/amazon

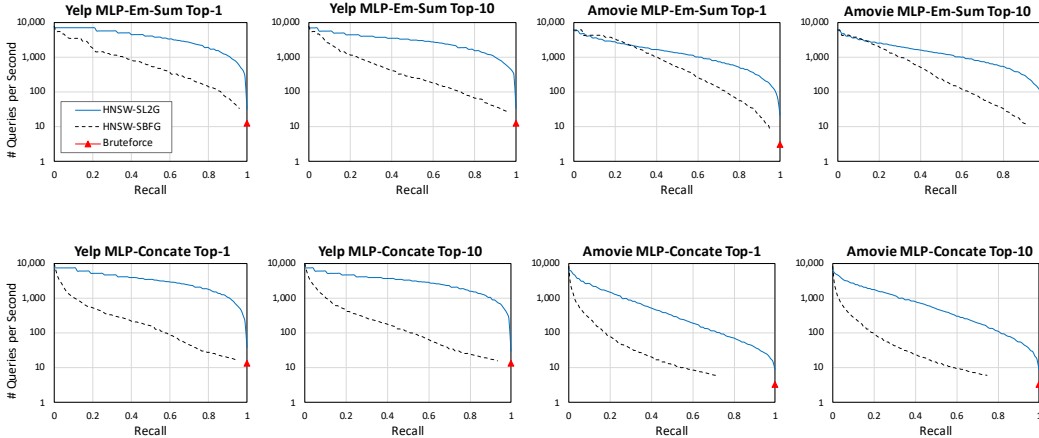

Figure 2: Experimental results for efficient ranking by Neural Network based ranking measures. The first row is for $f_{\text{MLP-Em-Sum}}$ and the second row is for $f_{\text{MLP-Concate}}$. Top-N means we only care about top N labels for the corresponding experiments. Top-1 and Top-10 results are shown here.

by the algorithm for HNSW. The graph built like this may be dramatically different from Delaunay graph w.r.t. $f(x, q)$. So there is no performance guarantee for HNSW-SBFG. Note that we restrict dimensions for users and items the same in MLP-Concate as mentioned above. In this way, HNSW-SBFG can be implemented. Otherwise, if the dimensions for users and items are different, $f(x, x)$ is invalid. Obviously, SL2G has no such limitations.

Experimental results for Top-1 and Top-5 labels are shown in Figure 2. Results for the symmetric $f_{\text{MLP-Em-Sum}}$ are corresponding to the first row of the figure. As can be seen, although SBFG can speed up the ranking process at some recall levels, it works much worse than the proposed SL2G. For the asymmetric $f_{\text{MLP-Concate}}$, with results shown in the second row of Figure 2, the superiority of SL2G over SBFG is more significant. The asymmetric characteristic poses challenges for the SBFG methodology but SL2G, which is proposed for general binary functional search, can overcome the challenges naturally. Overall, we can see that SL2G can speed up the searching process for Neural Network based measures tens of times comparing with the brute force ranking. For example, in the experiment of "Yelp MLP-EM-Sum Top-1" (i.e., the first figure of Figure 2), to get $99.00\%$ recall, SL2G can process more than 264 queries but brute force ranking can only process 12.29 queries on average. It is a more than 20 times improvement.

Note that the graph construction for SL2G (w.r.t. $\ell_2$ distance computations) is much more efficient than SBFG (w.r.t. complicate $f$ computations). For example, for the case of Yelp and $f_{\text{MLP-EM-Sum}}$ with $M = 16$ and $efConstruction = 100$, SBFG takes 668.32 seconds to construct the graph but SL2G only takes 1.41 seconds.

Since the space is limited, we put the experimental results for Top-50 and Top-100 labels (with similar trends) in Appendix D. Although the proposed method, SL2G, is designed for complicated matching measures, we can also apply it for classical measures, such as Inner Product. The study for Maximum Inner Product Search (MIPS) can be found in Appendix E.

## 5 CONCLUSION

In this paper, we introduce a vital task, Optimal Binary Functional Search, which generalizes the traditional Approximate Nearest Neighbor Search problem. A graph based approximate search method, Search on L2 Graph (SL2G), is proposed for OBFS. Approximate ranking by complicated relevance measures, such as Neural Network based ones, can be speed up significantly by the proposed method. As shown in theoretical analysis, SL2G approximates gradient descent in Euclidean space. Even for non-convex matching measures, SL2G guarantees that at least an approximate local optimal can be found based on some weak assumptions. In experiments, two Neural Networks for recommendation are selected as the matching measures to test SL2G. Results shows that SL2G can speed up the searching efficiency tens of times, comparing with brute force computations.

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

## A    PROOF FOR THEOREM 1

*Proof.* By the assumption on $f$, we have

$$\tilde{R}_i = \bigcap_{x \in N(x_i)} \{q \in Y : f(x_i, q) \geq f(x, q)\} \tag{4}$$

is connected and $R_i \cup \{q\} \subset \tilde{R}_i$. Hence we can define a path $c : [0, 1] \to Y$ such that $c(0) \in R_i$ and $c(1) = q$. For every $x_j \in S$, $f(x_j, c(0)) \leq f(x_i, c(0))$. If $f(x_j, c(1)) \geq f(x_i, c(1))$, then by intermediate value theorem, there exists $t \in [0, 1]$ such that $f(x_j, c(t)) = f(x_i, c(t))$. Hence $R_i \cap R_j \neq \emptyset$, and $x_j$ is a neighbor of $x_i$ on $G$. In this case, by equation 3, we must have $f(x_i, q) = f(x_j, q)$. Therefore, for $x_j \in S$, we have $f(x_i, q) \geq f(x_j, q)$. □

## B    A GUARANTEE BOUND FOR SL2G

**Proposition 1.** *Let $S$ follows Poisson point process with density function $\lambda : E \to \mathbb{R}_+$. We assume $\inf_{x \in E} \lambda(x) \geq \lambda^* > 0$, then the condition (b) in Theorem 2 is satisfied with probability at least $1 - C_r \exp(-\lambda^* V(r/2, d))$, where $C_r$ only depends on the shape of $E$ and $r$, and $V(r/2, d)$ is the volume of a $r/2$-ball in $\mathbb{R}^d$.*

*Proof.* Consider an open cover $\{B_x(r/2) : x \in E\}$ of $E$, and choose a finite subcover with minimum number of balls contained in $E$, say there are $C_r$ such balls, where $C_r$ only depends on the shape of $E$ and $r$. We denote the centers of these balls by $F = \{x_i : i = [C_r]\}$. We emphasize that we only consider the balls contained in $E$ in the subcover. For $x_0$ such that $B_{x_0}(r/2) \subset E$, let $\Lambda_{x_0} = \int_{B_{x_0}(r/2)} \lambda(x)dx \geq \lambda^* V(r/2, d)$, and $X \sim \text{Pois}(\Lambda_{x_0})$, then

$$\mathbb{P}(B_{x_0}(r/2) \cap S = \emptyset) = \mathbb{P}(X = 0) = \exp(-\Lambda_{x_0}) \leq \exp(-\lambda^* V(r/2, d)),$$

Considering all balls with centers in $F$, we have

$$\mathbb{P}(\exists i \in [C_d], B_{x_i}(r/2) \cap S = \emptyset) \leq C_r \exp(-\lambda^* V(r/2, d)).$$

Let $p := 1 - C_r \exp(-\lambda^* V(r/2, d))$. For every ball $B_{x'}(r) \subset E$, there exists $x_i \in F$ such that $x' \in B_{x_i}(r/2) \subset B_{x'}(r)$. Hence with probability at least $p$, every ball $B_{x'}(r) \subset E$ contains a point in $S$. □

Assuming $r$ and $E$ are fixed, as we obtain a larger dataset, in other words, when minimum density of Poisson point process on $E$ increases with the same rate as $n$, i.e., $\lambda \propto n$, the failing probability in Proposition 1 converges to zero in an exponential rate.

## C    ALGORITHMS

## D    MORE EXPERIMENTAL RESULTS FOR $NN$ BASED BINARY FUNCTIONALS

More experimental results for Neural Network based binary functionals are shown in Figure 3. As can be seen , results for Top-50 and Top-100 labels have similar trends with those for Top-1 and Top-10. The superiority of SL2G is even more significant when we increase the label amount.

---

**Algorithm 2** Binary Functional Graph Construction for SBFG

---

  1: **input:** binary functional $f$, data set $S$ and maximum vertex degree $M$
  2: Initialize graph $G = \emptyset$
  3: **for** each $x_i$ in $S$ **do**
  4:      Find $M$ vertices $\{x_j\}$ on $G$ with largest values in $f(x_i, x_j)$          ▷ only for symmetric $f$
  5:      Construct $M$ undirected edges in $G$ for each pair of $x_i$ and $x_j$
  6: **end for**
  7: **output:** graph $G$

---

---

**Algorithm 3** Greedy Search on Graph for both SBFG and SL2G

---

  1: **input:** a graph $G$, a binary functional $f$, a query $q$ and a randomly selected enter point $v^*$
  2: Initialize return vertex $v = v^*$ and $v_{pre} = null$
  3: **while** $v$ is not the same with $v_{pre}$ **do**
  4:      $v_{pre} = v$
  5:      **for** each adjacent vertex $v_i$ of $v$ **do**
  6:          **if** $v_i$ is not visited and $f(v_i, q) > f(v, q)$ **then**
  7:              $v = v_i$
  8:          **end if**
  9:      **end for**
10: **end while**
11: **output:** vertex $v$

---

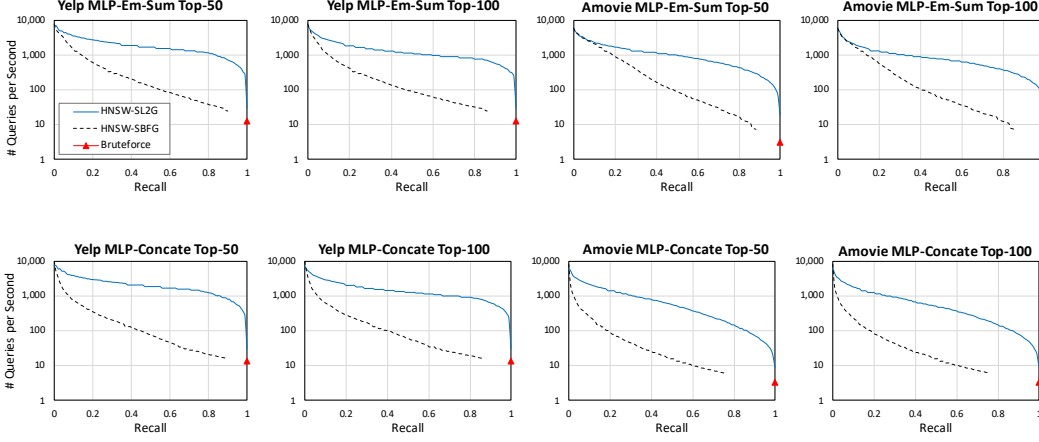

Figure 3: Experimental results for efficient ranking by Neural Network based ranking measures corresponding to cases of Top-50 and Top-100 labels.

# E    STUDY ON MAXIMUM INNER PRODUCT SEARCH (MIPS)

In this section, we perform an empirical study on the MIPS problem based on the theory of this paper. MIPS extracts many attentions because of its broad application scenarios. For instance, Matrix Factorization based recommendation methods naturally pose a MIPS problem. Since Inner Product is non-metric (i.e., does not hold triangle inequality), MIPS is a different problem from traditional ANNS on unnormalized searching data. Intuitively, even Inner Product is a non-metric measure, its simplicity makes approximating Delaunay graphs w.r.t. Inner Product affordable so that SBFG would work well on MIPS. We utilize HNSW to approximately construct Delaunay graphs w.r.t. Inner Product and test SBFG for MIPS. SL2G can also be used MIPS with HNSW. Different from complicated measures, there are some previous methods for MIPS, besides of SBFG and SL2G. We introduce two more state-of-the-art MIPS methods as baselines here, which are both based on the methods of transforming MIPS to ANNS in metric spaces by pre-processing indexed data and queries asymmetricly (Shrivastava & Li, 2014; Bachrach et al., 2014). The first one is adding the processing wrapper on Sign $p$-Stable Projection Hash (Datar et al., 2004) (**SSPH-Wrapper**) and the second on is processing data before exploiting HNSW with $\ell_2$ distance as the measure (**HNSW-Wrapper**). Specifically, we process the data as below. We note the query as $q$, the indexed data as $x_i$ and $\Phi = max_i ||x_i||$. The wrapper is defined as:

$$P(x) = [x/\Phi; \sqrt{1 - ||x||^2/\Phi^2}], \tag{5}$$
$$Q(q) = [q; 0]. \tag{6}$$

It is not difficult to proof that searching on the new data by $\ell_2$ distance or angular distance is equal to search on the original data by Inner Product.

$$\underset{i}{\operatorname{argmin}}(P_i(x) - Q(q))^2 = \underset{i}{\operatorname{argmax}} \left( x_i \cdot q - \frac{1}{2} \left( ||Q(q)||^2 + \Phi^2 \right) \right) = \underset{i}{\operatorname{argmax}} x_i \cdot q. \tag{7}$$

$$\underset{i}{\operatorname{argmax}} \frac{P_i(x) \cdot Q(q)}{||P_i(x)|| ||Q(q)||} = \underset{i}{\operatorname{argmax}} \frac{x_i \cdot q}{||Q(q)||\Phi} = \underset{i}{\operatorname{argmax}} x_i \cdot q. \tag{8}$$

Note that the main purpose of this section is to test the applicability of the propose SL2G on simple relevant measures (i.e., Inner Product). MIPS is not the focus of this paper, so we do not involve and compare with more state-of-the-art MIPS methods, such as Guo et al. (2016) and Yu et al. (2017).

To get the vectors for the MIPS testing, we use two different recommendation models. The first one is Matrix Factorization (**MF**) introduced in Hu et al. (2008) and we utilize the implement in Ben Frederickson et al.. The vector data produced by this method will be referred as **Yelp MF** and **Amovie MF**. The second one is a Neural Network based Recommendation model, Deep Matrix Factorization (**DMF**) (Xue et al., 2017). Correspondingly, we will have another two vector datasets: **Yelp DMF** and **Amovie DMF**. The length for all vectors is set as $64$. Note that vectors produced by MF and DMF are quite different. Vectors produced by MF contains both positive and negative real numbers but vectors from DMF are sparse and only contain negative numbers because of the RELU active function.

Experimental Results are shown in Figure 4. As can been seen, graph based methods show their great superiority in performance, comparing with other methods (e.g., SSPH-Wrapper), which is consistent with the previous study (Aumüller et al., 2017). For the three graph based methods, it is interesting that HNSW-SBFG works best for Top-1 labels on various data sets but works worse and worse when we consider more labels. For Top-100 labels, it works much worse than the other two methods. The reason is that searching on Delaunay graphs w.r.t. Inner Product, some data points may be never returned. They are always "represented" by the "leader" data point in the corresponding Voronoi cells. For example, $S = \{(0, 1), (0, 0), (0.8, 0.1), (1, 0)\}$. Then the Delaunay graph w.r.t. $S$ and $f$ only connects $(0, 1), (0, 0)$ and $(1, 0)$. Suppose our task is to return top-2 results and $q = (2, 0)$ comes in, the greedy search will never return the second best result $(0.8, 0.1)$. Conversely, HNSW-Wrapper and HNSW-SL2G work consistently well by varying the amount of top labels and show their stable performance.

Although HNSW-SL2G is not specifically designed for the MIPS problem, it works very closely to HNSW-Wrapper, which has theoretical guarantees for MIPS. That demonstrates the applicability of SL2G for various binary functionals.

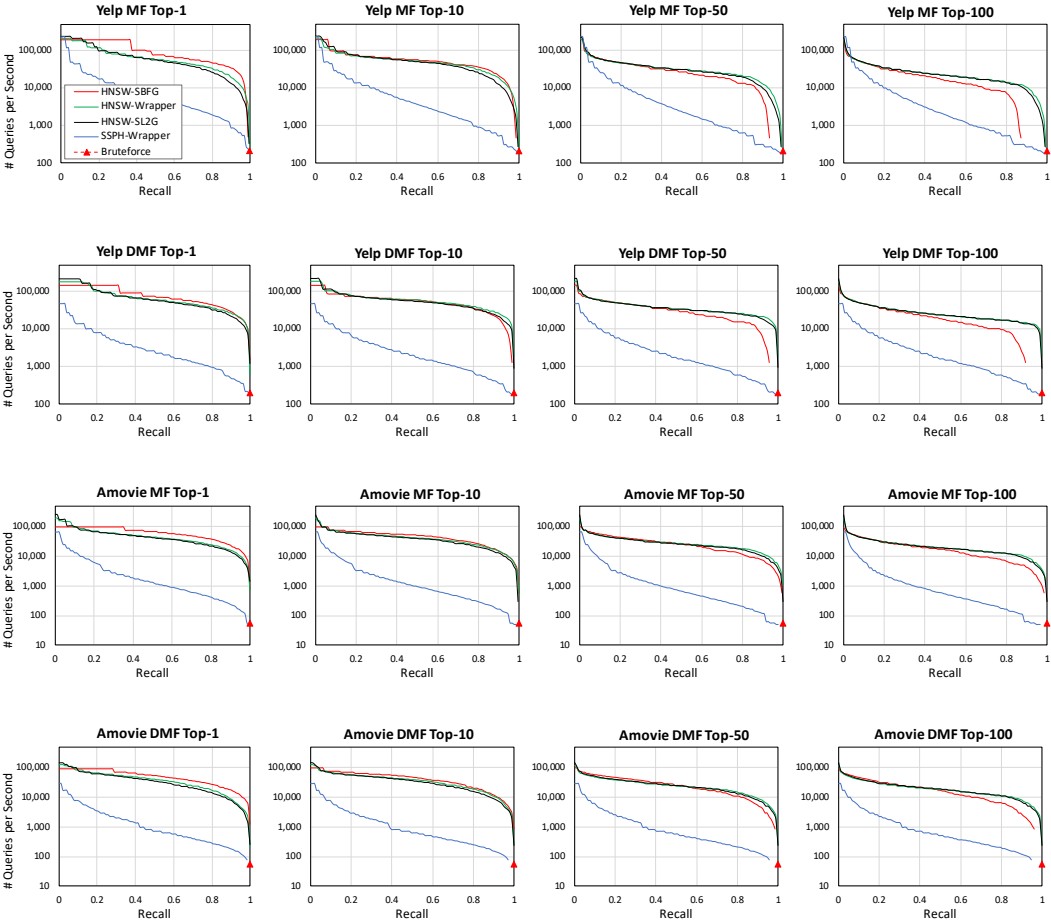

Figure 4: MIPS experimental results. The first row is for Yelp data learned by MF. The second row is for Amovie data learned by MF. The third row is for Yelp data learned by DMF. The fourth row is for Amovie data learned by DMF. Columns are for results by varying the amount of top labels (i.e., Top-1, 10, 50 and 100).

