# OpenReview forum: "Fast Binary Functional Search on Graph"
_ICLR.cc/2019/Conference_

### Official Review · AnonReviewer1 · 2018-10-23
**Promising novel idea; needs further clarification and development**

**Rating:** 5
**Confidence:** 5

**Review:**

Post-rebuttal
------------------
I have read the rebuttal and I better understand the paper. Given that, I am going to raise my rating by one point for the following reason:
- The manuscript presents a novel solution to a general problem and it is a valid solution. However, the solution is somewhat obvious, which is not necessarily a bad thing, which is why I am raising my rating by a point. However, an easy solution like the one proposed in the manuscript means that OBFS considered in this manuscript is not as general as the authors let on -- there is an implicit assumption that f(x_i, q) is close to f(x_j, q) if x_i is close to x_j.
- While the authors answered a lot of my clarification questions, the manuscript seems still a little hard to parse and can be significantly improved for easier reading and understanding.

=========================================
Pros
-------
[Originality/Significance] The manuscript focuses on a very general and important problem and proposes a scheme to solve this general problem. The authors present some theoretical and empirical results to demonstrate the utility of the proposed scheme.

Limitations
----------------
[Clarity] While the problem being addressing is extremely important, and the proposed solution seems reasonable, the manuscript is really hard to follow. For example, Definition 3 and Theorem 1 are extremely hard to understand.

[Clarity/Significance] Moreover, I feel that the authors should be more precise in pointing out why current graph based search algorithms are just not trivially applicable to OBFS. The nature of the approximate Delaunay graph is that it can be built for any given similarity function (the level of approximation obviously depends on the similarity function, but that is an existing issue with graph-based methods). Given the graph, I do not understand why the basic search algorithm on this similarity graph would not be an approximate solution to OBFS. Hence I believe the authors need to clarify why the existing graph based algorithms do not directly translate.

[Significance] While Definition 1 considers topological spaces, SL2G is assuming that X and (maybe) Y are in R^d (for different values of d). So does that mean that SL2G does not solve the general OBFS?

[Significance/Correctness/Clarity] The assumptions in Theorem 2 (as well as the supporting Proposition 1 in Appendix B) seems quite unreasonable. In moderately high dimensional X, doesn't the curse of dimensionality imply that this condition will not hold in most case? In there any reason why/how this would be circumvented? Moreover, in Proposition 1 (in Appendix B), the quantity C_r needs to be precisely defined since it could in general be exponential in the number of dimensions. Also, the assumption in Proposition 1 where \lambda^* > 0 is fairly strong in high dimensional data since data gets really sparse in high dimensions. Finally, the last step in Proposition 1 (where the failure probability obtained from the union bound is connected to condition (b) in Theorem 2) is not clear at all -- it is not apparent how E and F related to S and how p relates to every ball containing a point in S. This is a very important step and needs better exposition.

[Clarity/Significance] I am unable to understand the baseline HNSW-SBFG (or the motivation for it) in the empirical section. It would be good to clarify this.


General comments
---------------------------
[Significance] Finally, I believe that it would be good to see a connection between the success of SL2G to relationship between |f(x1, q) - f(x2, q)| and ||x1 - x2 ||_2 since the author emphasize that the proposed scheme can be seen as "gradient descent in Euclidean space" (although the authors would need to also precisely explain what they mean by that statement).

[Originality] Some related work that the authors should position their proposed problem/solution against:
- There is some work on "max-kernel search" which can perform similarity search with general notions of similarity (than just Euclidean metrics).
- There is some work on search with Bregman divergences which handle asymmetric similarity functions and also incorporate notions of gradient descent over convex sets.

Minor comments/typos
---------------------------------
- The authors should present the precise SL2G algorithm given the graph in the manuscript.
- l^2 --> \ell_2
- gradient decent --> gradient descent
- Table 1, f(q, x) --> f(x, q)

---

> ### Author Response · Authors · 2018-11-15
> **Responses to the Comment1&2**
>
> 1. While the problem being addressing is extremely important, and the proposed solution seems reasonable, the manuscript is really hard to follow. For example, Definition 3 and Theorem 1 are extremely hard to understand.
>
> [Response] Thanks for your comments. We will add more explanations for the theory part and make it easier to access. Specifically, although the problem is in an asymmetric setting, readers can still assume f(x,y) = -|x-y| as a typical example to understand the definitions and theorem. For example, assuming f is the negative l2-norm, then definition 3 means we will connect two data points in the Delaunay graph if the Voronoi cell is “adjacent” to each other. Here, adjacency means their boundary has nonempty intersection. Theorem 1 means that, for an arbitrary query, a greedy search on Delaunay graph with any initial point can find the nearest neighbor of the query.
>
> 2. [Clarity/Significance] Moreover, I feel that the authors should be more precise in pointing out why the current graph-based search algorithms are just not trivially applicable to OBFS. …
>
> [Response] Thank you for this comment. We are going to assume "basic search algorithm on similarity graph" indicates the previous search on graph methods, such as HNSW or Bregman ball tree (you mention in a later comment). These algorithms require f(x,y) defined on the product of two identical spaces. OBFS is much more general and does not have such an assumption.
>
> Suppose we still assume x and y are from the same space and plug in f as a "similarity function" in HNSW, which is exactly the baseline, HNSW-SBFG, we used in experiments. Particularly, in the recommendation-system scenario, we embed users and items in the same Euclidean space. As shown in the experimental results on page 8, the performance of HNSW-SBFG is much poorer than HNSW-SL2G. We believe original HNSW or any other existing similarity graph based algorithms require f performs like a similarity function. A well behaved f in recommendation system should not measure the similarity between user and item.
>
> It is also worth to mention that, although we provide guarantees for SBFG, but most of general f's, e.g., neural networks, does not satisfy the condition in Theorem 1.

---

> ### Author Response · Authors · 2018-11-15
> **Responses to the Comment 3, 4&5**
>
> 3. [Significance] While Definition 1 considers topological spaces, SL2G is assuming that X and (maybe) Y are in $R^d$ (for different values of d). So does that mean that SL2G does not solve the general OBFS?
>
> [Response] Thanks for pointing this out. The answer is “no”, it only solves OBFS when X and Y are subsets of Euclidean spaces. Actually, we are usually interested in Euclidean spaces or its subsets in real applications, as mentioned below Definition 1.
>
> 4. [Significance/Correctness/Clarity] The assumptions in Theorem 2 (as well as the supporting Proposition 1 in Appendix B) seems quite unreasonable. In moderately high dimensional X, doesn't the curse of dimensionality imply that this condition will not hold in most case? …
>
> [Response]Thank you for reading the proposition and theorem very carefully.
> We consider an asymptotic setting that the number of data points growing to infinity and the dimension of X is fixed. When the region E is fixed, \lambda^* is proportional to the number of data points, so it goes to infinity.
> Of course, one can consider a high dimensional setting when n and d increase simultaneously. However, C_r is still not critical since it is not in the exponent. In the failing probability formula, the volume of r/2 ball depends on d and plays a much more important role than C_r when d increases. If we hope the failing probability still goes to 0, then we should require log \lambda^* is much greater than d log d.
> About the implication from Proposition 1 to condition (b) in Theorem 2, it is a simple geometry property. We recall that F is a set of centers of open r/2-balls whose union covers E. For a fixed open r-ball, say B, its center is covered by at least one open r/2-ball with the center in F. This open r/2-ball is contained in B by triangle inequality. The open r/2-ball contains at least one data point, which also belongs to B. This implies every open r-ball contains a data point, which is the assumption (b) in Theorem 2. We believe our proof is mathematically correct and clear.
> We believe this is a nontrivial result. As the number of data points increases, we have more “bad” data points. Here, “bad” data point means it is far away from the local optimum of $f$, but it is a local optimum in greedy search on the graph. The theorem and proposition show that even if we have more bad data points, the failure probability of the greedy search still goes to 0.
>
> 5. [Clarity/Significance] I am unable to understand the baseline HNSW-SBFG (or the motivation for it) in the empirical section. It would be good to clarify this.
>
> [Answer] HNSW-SBFG is quite similar to the original HNSW. We just replace the metric measure in HNSW, such as l2 or cosine, with the focusing search binary functional f. Beyond that, the graph construction and greedy search approaches of HNSW-SBFG are same as the original HNSW. Note that, to let HNSW-SBFG be applicable, we set X and Y in the same space (both 64-dimensional). In this way, f(x_i,x_j) will output a value no matter f is symmetrical (e.g., MLP-Em-Sum) or asymmetrical (e.g., MLP-Concate). If f is asymmetrical, f(x_i,x_j) is problematic actually. That why HNSW-SBFG works even worse on MLP-Concate datasets. If X and Y have different dimensions, HNSW-SBFG will be not applicable.

---

> ### Author Response · Authors · 2018-11-15
> **Responses to the Comment 6, 7&8**
>
> 6. [Significance] Finally, I believe that it would be good to see a connection between the success of SL2G to relationship between |f(x1, q) - f(x2, q)| and |x1 - x2 |_2 since the author emphasize that the proposed scheme can be seen as "gradient descent in Euclidean space" (although the authors would need to also precisely explain what they mean by that statement).
>
> [Response] As we mentioned in the paragraph after Theorem 2, the success and accuracy of our algorithm depend on the radius of curvature of the level sets. We believe |f(x_1, q) - f(x_2, q)| and |x_1 - x_2 |_2 together with the density of dataset might affect the speed of convergence, which we have not covered in this paper.
>
> 7. [Originality] Some related work that the authors should position their proposed problem/solution against…
>
> [Response] Thank you for pointing out these related works. First of all, we would like to emphasize that our work is very different from similarity search, so most of the existing methods in this field does not apply to our problem.
> ``Max-kernel search" is defined on a Hilbert space, so it has to be symmetric. Bregman divergence does not have to be symmetric, but both variables must come from the same convex set. Even if we apply Bregman ball tree to a similarity search problem, we do not think it performs well on finding top-10 nearest neighbors.
> We will analyze these works in our updated manuscript.
>
> 8. The authors should present the precise SL2G algorithm given the graph in the manuscript.
>
> [Response] Thanks for your suggestion. To make the manuscript more self-contained, we will list the algorithms (graph construction and greedy search) in the Appendix, although they are common algorithms for search on graph methods.
>
> Finally, we really appreciate your time and detailed comments.

---

> ### Comment · AnonReviewer1 · 2018-11-29
> **Response to author rebuttal**
>
> Thank you for the detailed rebuttal. It has been very helpful for me to better understand the proposed solution and its utility.
>
> What I meant with "basic search algorithm" was that for any functional f(x_i, q), a graph would be built on S on some distance (or inverse similarity) function d(x_i, x_j) such that small d(x_i, x_j) implies small |f(x_i, q) - f(x_j, q)| and large d(x_i, x_j) implies large |f(x_i, q) - f(x_j, q)|. Then the search would just proceed with f for any query q. But now I see that this is exactly your proposed SL2G where you fix your d(x_i, x_j) to be the \ell_2 distance. So essentially the guarantee for accuracy is dependent on the Lipschitz constant of the function f(\cdot, q) = f_q(\cdot) for a fixed q.
>
> OBFS is definitely more general definition than Bregman search and Max-kernel search but it is just a definition. This paper provides a solution SL2G to OBFS (defined on Euclidean spaces) but SL2G has no concrete correctness guarantees to the best of my understanding, especially since it is very very hard to build exact Delaunay graph (even for incomplete "local" ones) and the guarantees in this manuscript do not account for this approximation. On the other hand, Bregman search and max-kernel search provide correctness guarantees (using the structure of the problem). But that may be a shortcoming of graph-based search algorithms in general, not SL2G in particular.

---

### Official Review · AnonReviewer2 · 2018-11-05
**Fast Binary Functional Search on Graph**

**Rating:** 4
**Confidence:** 4

**Review:**

This work extends the approximate nearest neighbor search (ANNS) algorithm to a more general setting. Instead of search with a "separable" similarity measure, the authors propose Optimal Binary Functional Search (OBFS), where the scoring function f() is in general non-separable. The exact construction of the Binary Function Graph wrt f() and X is computationally expensive. The specific approximate algorithm of OBFS proposed in the paper is to:
1) First construct an L2 Delaunay graph for based on the dataset X only and;
2) Perform greedy search with the L2 Delaunay graph.

The authors also discuss various conditions under which, the approximation method can achieve close to optimal value.

Some of the concerns I have with this work:

1) The authors do not demonstrate sufficient value of performing approximation in this specific fashion. For instance, in  Theorem 2, the authors start with the concavity assumption of the scoring function f(). Then it is natural to apply a gradient ascent method on the neighborhood graph. And the authors did not quantitatively or qualitatively justify their specific approach.

2) Lately, numerous publications have shown that distilled models can achieve very high quality and render scoring function separable. The authors should at least compare their method against distillation and Maximum Inner Product Search based approaches.

Overall, this research direction is interesting, but this specific work falls short for a publication at ICLR.

---

> ### Author Response · Authors · 2018-11-15
> **Responses to the comments**
>
> The authors do not demonstrate sufficient value of performing approximation in this specific fashion. For instance, in Theorem 2, the authors start with the concavity assumption of the scoring function f(). Then it is natural to apply a gradient ascent method on the neighborhood graph. And the authors did not quantitatively or qualitatively justify their specific approach.
>
> [Response] Thanks for your comments. Nodes on the neighborhood graph are discrete points in the space. Searching on the neighborhood graph is quite different from gradient descent in the continuous space. That is why we try to figure out the conditions in which the proposed method will work well. To the best of our knowledge, this is the first work discusses this point. We provided the theoretical analysis and empirical experiments for the proposed approach.
>
> Lately, numerous publications have shown that distilled models can achieve very high quality and render scoring function separable. The authors should at least compare their method against distillation and Maximum Inner Product Search based approaches.
>
> [Response] For related distilled models, could you specify the particular papers? Thanks.
>
> The MIPS problem is a special case of the Binary Functional Search problem. Although the proposed method (SL2G) is not designed for the MIPS problem but for more complex searching measures, it can be applied for MIPS, the corresponding empirical study can be found in Appendix D.

---

> > ### Comment · AnonReviewer2 · 2018-11-29
> > **Not enough for me to change my review.**
> >
> > Thanks for the response. For general distillation, please refer to the original paper:
> > https://arxiv.org/abs/1503.02531
> >
> > As to your specific case, I see that you are using recommendation datasets.
> > Please consider using factorized model with side information.

---

### Meta-Review · Area_Chair1 · 2018-12-14
**Rejection: good Paper but still require further improvement**

**Confidence:** 4
**Recommendation:** Reject

**Metareview:**

This paper proposes an Optimal Binary Functional Search (OBFS) algorithm for searching with general score functions, which generalizes the standard similarity measures based on Euclidean distances. This yields an extension of the classical approximate nearest neighbor search (ANNS). As observed by the reviewers, this work targets an important research direction. Unfortunately, the reviewers raised several concerns regarding the clarity and significance of the work. The authors provided a good rebuttal and addressed some concerns, but not to the degree that reviewers think it passes the bar of ICLR. We encourage the authors to further improve the work to address the key concerns.